# The PARP Inhibitor Olaparib Modulates the Transcriptional Regulatory Networks of Long Non-Coding RNAs during Vasculogenic Mimicry

**DOI:** 10.3390/cells9122690

**Published:** 2020-12-15

**Authors:** Mónica Fernández-Cortés, Eduardo Andrés-León, Francisco Javier Oliver

**Affiliations:** Instituto de Parasitología y Biomedicina López Neyra, CSIC, CIBERONC, 18016 Granada, Spain; monica.fernandez@ipb.csic.es (M.F.-C.); eduardo.andres@csic.es (E.A.-L.)

**Keywords:** vasculogenic mimicry, lncRNA integrative analysis, uveal melanoma, olaparib, PARP inhibitors

## Abstract

In highly metastatic tumors, vasculogenic mimicry (VM) involves the acquisition by tumor cells of endothelial-like traits. Poly-(ADP-ribose) polymerase (PARP) inhibitors are currently used against tumors displaying BRCA1/2-dependent deficient homologous recombination, and they may have antimetastatic activity. Long non-coding RNAs (lncRNAs) are emerging as key species-specific regulators of cellular and disease processes. To evaluate the impact of olaparib treatment in the context of non-coding RNA, we have analyzed the expression of lncRNA after performing unbiased whole-transcriptome profiling of human uveal melanoma cells cultured to form VM. RNAseq revealed that the non-coding transcriptomic landscape differed between olaparib-treated and non-treated cells: olaparib significantly modulated the expression of 20 lncRNAs, 11 lncRNAs being upregulated, and 9 downregulated. We subjected the data to different bioinformatics tools and analysis in public databases. We found that copy-number variation alterations in some olaparib-modulated lncRNAs had a statistically significant correlation with alterations in some key tumor suppressor genes. Furthermore, the lncRNAs that were modulated by olaparib appeared to be regulated by common transcription factors: ETS1 had high-score binding sites in the promoters of all olaparib upregulated lncRNAs, while MZF1, RHOXF1 and NR2C2 had high-score binding sites in the promoters of all olaparib downregulated lncRNAs. Finally, we predicted that olaparib-modulated lncRNAs could further regulate several transcription factors and their subsequent target genes in melanoma, suggesting that olaparib may trigger a major shift in gene expression mediated by the regulation lncRNA. Globally, olaparib changed the lncRNA expression landscape during VM affecting angiogenesis-related genes.

## 1. Introduction

While less than 2% of the human genome is transcribed into genes coding for proteins, most of the human transcriptome (around 80%) is transcribed into non-coding RNA. These non-translated regions of the genome were wrongly viewed as “junk DNA” for a long time. In recent years there is an increasing interest in the field of non-coding RNAs, including long non-coding RNAs (lncRNAs). LncRNAs comprise various RNA species longer than 200 nucleotides in size, which are not transcribed into proteins. They have been classified into sense lncRNAs, antisense lncRNAs, intergenic lncRNAs, intronic lncRNAs and bidirectional lncRNAs [1,2]. LncRNAs play important roles in the development and progression of various human cancers through different mechanisms. They have been defined as “key players” in tumor development due to their ability to regulate pivotal processes like cell differentiation and cell death, as well as their involvement in pathophysiological processes [3]. *Cis*-acting lncRNAs have been demonstrated to activate, repress or otherwise modulate the expression of their surrounding genes. Within gene regulatory networks, lncRNAs can act alongside *cis* factors, such as DNA regulatory elements and epigenetic modifications, as well as *trans* factors, such as transcription factors and small non-coding RNA. At the epigenetic level, lncRNAs can regulate chromatin structure by recruiting and/or binding histone-modifying enzymes and DNA methyltransferases [4]. At the transcriptional level, lncRNAs may interact with specific transcription factors, serving as a scaffold or, on the contrary, occupying DNA-recognition elements and preventing their binding to DNA. In both cases, lncRNA interaction can change the expression of target genes. Furthermore, lncRNAs can also modulate expression in a post-transcriptional manner, namely affecting splicing or initiation of translation [3].

Early during their development, tumors undergo the angiogenic switch, inducing the growth of new blood vessels to facilitate the supply of oxygen and nutrients and enable the elimination of metabolic waste. Moreover, these blood vessels promote metastatic spread. Angiogenesis is the most widely investigated mode of tumor neovascularization, though alternative mechanisms, such as vasculogenic mimicry (VM), have been gaining attention in recent years. VM describes the generation of perfusion pathways in tumors by highly aggressive and genetically deregulated tumor cells, independently of angiogenesis or pre-existing normal blood vessels. The presence of VM is reportedly correlated with increased metastasis, poor prognosis and decreased survival in cancer patients. Moreover, VM is associated with resistance to conventional anti-angiogenic therapies [5].

LncRNAs have been proposed as important regulators of tumor angiogenesis, affecting tumor-associated cells, oncogenic pathways or binding to other RNA transcripts [6]. For instance, lncRNA HIF2PUT acts as a direct upstream promoter inducing EPAS1 (HIF2A) expression in colorectal cancer and consequently increasing angiogenesis; another example is the lncRNA PVT1, which stabilizes STAT3 and therefore indirectly increases VEGF expression in gastric cancer [7]. On the contrary, lncRNA hypoxia-inducible factor-1 alpha subunit antisense RNA 2 (HIF-1A-AS2) forms a complex with HIF-1α-mRNA, leading to a negative feedback loop, blocking angiogenesis in non-tumor tissue [8]. Several reports have shown a role for lncRNAs in VM as well. MALAT1, one of the most studied lncRNAs due to its role in various cancer types, was shown to promote VM in lung [9] and gastric [10] cancer. LINC00312 induced VM in lung cancer cells by modulating the transcription factor YBX1 [11]. In glioma cells, SNHG20 promotes the degradation of the transcription factor FOXK1, which results in increased VM [12]. In most of these examples, the ability to regulate VM is mediated by changes in the expression of the vascular endothelial (VE-)cadherin, a well-known marker of VM [13].

Poly-(ADP-ribose) polymerase (PARP) inhibitors are used against BRCA1/2-mutant breast and ovarian cancer, although their use could be extended to other oncological settings, including malignant transformations. In spite of the extended use of PARP inhibitors as antitumoral agents, no systematic study has been performed to analyze lncRNA modulation by olaparib in any cancer setting. In a previous study, we showed that PARP inhibition altered the ability of melanoma to undergo VM, interfering with the expression and phosphorylation of VE-cadherin [14]. Here we analyzed the differential expression profiles of lncRNA in olaparib-treated uveal melanoma (UM) in a 3D model of VM and investigated the relationship between the lncRNA expression after olaparib treatment to provide a theoretical basis for the selection of new targets against abnormal tumor angiogenesis. We identified a group of differentially expressed lncRNAs affected by olaparib during VM development in malignant UM cells, being FLG-AS1 the most upregulated and RP11−706O15.5 the most downregulated. Analysis of transcription factor binding to promoters of olaparib-modulated lncRNAs revealed a common set of transcription factors that might be involved in olaparib-dependent lncRNA upregulation (ETS1) or downregulation (MZF1, RHOXF1 and NR2C2). Moreover, we show that olaparib-modulated lncRNAs might, in turn, regulate a number of other transcription factors and their target genes.

Overall, we propose that olaparib treatment may trigger a major transcriptome shift mediated by the regulation of the lncRNA landscape.

## 2. Methods

### 2.1. Cell Culture and RNA Extraction

Metastatic UM cell line Mum2B was grown in RPMI medium supplemented with 10% fetal bovine serum, 2 mM of L-glutamine and 1% penicillin/streptomycin, cultured at 37 °C and 5% CO_2_.

For RNA extraction, 35 mm culture dishes were coated with 750 µL Matrigel (Corning), allowed to solidify at 37 °C for 30 min. 3.8 × 10^5^ Mum2B cells were cultured on Matrigel and treated with DMSO or olaparib 5 µM. After 24 h, cells were harvested with a cell recovery solution (Corning), following the manufacturer’s recommendations. RNA was subsequently extracted with RNeasy Mini Kit from Qiagen (Hilden, Germany).

### 2.2. RNA Sequencing

Library preparation and Illumina sequencing were carried out at the IPBLN Genomics Facility (CSIC, Granada, Spain). Total RNA quality was verified by Bioanalyzer RNA 6000 Nanochip electrophoresis (Agilent Technologies, Santa Clara, CA, USA). Every RNA sample showed a RIN value above 9.6. RNA-seq libraries were prepared using TruSeq stranded mRNA kit (Illumina^®^, San Diego, CA, USA) from 800 ng of input total RNA. Quality and size distribution of PCR-enriched libraries were validated through Bioanalyzer high sensitivity DNA assay, and concentration was measured on the Qubit^®^ fluorometer (Thermo Scientific, Waltham, MA, USA). Finally, two types of samples, Olaparib− (control samples) and Olaparib+, were sequenced. From each type of sample, three biological replicates were made, giving a total of 6 libraries. These were pooled in an equimolecular manner and then diluted and denatured as recommended by Illumina NextSeq 500 library preparation guide. The 75 × 2 nt paired-end sequencing was conducted on a NextSeq 500 sequencer, producing 29,938,369 paired reads per sample on average.

### 2.3. Data Analysis

Transcriptomic samples were analyzed using the miARma-Seq pipeline [15]. Initially, raw sequence data were submitted to evaluation by means of FastQC software, which provides an in-depth report of the quality of the reads (Andrews, 2010; http://www.bioinformatics.babraham.ac.uk/projects/fastqc). Afterward, the number of reads per sample was homogenized using Seqtk software (Li, 2013). Subsequently, after sample filtering and trimming, we obtained an average of 2,864,540 fragments per sample with a mean of 49% in GC content. In the second step, miARma-Seq aligns all processed and quality filtered sequences using HISAT2 [16], resulting in 95.66% of properly aligned reads. For that purpose, we used the Homo sapiens Gencode version 26, genome-build: GRCh38.p10. In order to obtain their expression values, featureCounts software was used for assigning sequence reads to genes [17]. The reference gene and lncRNA annotation was also obtained from Gencode from the same assembly and genome build.

### 2.4. Differential Expression for mRNA and lncRNA

To perform the differential expression analysis, edgeR package was used [18]. Low expressed genes and lncRNAs were eliminated, and the remaining were normalized by the trimmed mean of M-values (TMM) method [19]. Counts per million (CPM) and log2-counts per million (log-CPM) were used for exploratory plots [18] to check the consistency of the replicates. Furthermore, reads per kilo base per million mapped reads (RPKM), was calculated per gene on each sample.

Principal component analysis (PCA) and hierarchical clustering of normalized samples were used to explore data and to get a general overview of the similarity of RNA-sequencing samples [20,21].

After this, differently expressed genes (DEG) and lncRNAs were calculated between untreated and olaparib-treated samples (false discovery rate (FDR) value < 0.05). To assess the change in expression of a lncRNA or a gene between the types of samples, the log2FC was provided. In order to visualize the most relevant genes in each comparison, a volcano plot is shown.

### 2.5. LncRNA and mRNA Expression Integration

LncRNA and mRNA RPKM’s were used to evaluate expression correlation among both types of RNAs. MatrixEQTL was used for this purpose [22]. This tool allows studying the correlation of expression between entities that share close genomic regions, 100 kbp in our case. MatrixEQTL was used with the default parameters, and all those relationships with a *p* < 0.01 were selected as significant.

### 2.6. Olaparib-Modulated lncRNAs in Cancer

The alteration status of olaparib-modulated lncRNAs in cancer samples was explored using the cBioPortal database [23], selecting The Cancer Genome Atlas (TCGA) PanCancer Atlas Studies. Unfortunately, not all information was available in this database for all lncRNAs. However, we could obtain some data for alteration frequency across different cancer types, mutual exclusivity of the different alterations, co-occurring mutations and overall survival. In addition, we investigated co-expression with PARP1/2 in the UM and the skin cutaneous melanoma (SKCM) TCGA PanCancer Atlas cohorts.

### 2.7. Transcription Factor Binding Sites Analysis

For every lncRNA identified as differentially regulated by olaparib, we defined an estimated promoter region between −1000 bp and +100 bp from the transcription start site. Each promoter was analyzed for transcription factor binding sites using the JASPAR database (http://jaspar.genereg.net/) [24], setting a relative profile score threshold of 95%.

### 2.8. Predicting lncRNA-Mediated Transcription Changes

We studied the potential effect of lncRNAs in modulating global gene expression using the database lncMAP (LncRNA Modulator Atlas in Pan-cancer; http://www.bio-bigdata.com/LncMAP) [25], which offers information about lncRNA-mediated transcription perturbations in different cancer types, listing lncRNA- transcription factor- gene triplets where the lncRNA can alter the activity of the transcription factor resulting in a change in gene expression. We searched each olaparib-modulated lncRNA in the SKCM cohort. Unfortunately, not all lncRNAs were available in this database. For those that were, lncMAP provided a list of transcription factors and downstream genes that were predicted to be regulated by each lncRNA, ranking lncRNA-transcription factor- gene triplets by FDR. Using this data, we made a graphic representation of the 30 most significant triplets predicted for each lncRNA, connecting every lncRNA (that we represented as a red square) to each transcription factor (orange triangle) that it was predicted to modulate, and further connecting every transcription factor to each gene (yellow circle) whose expression was predicted to change in response to the lncRNA-mediated transcription factor modulation.

We then searched for all predicted genes in our RNAseq data. Genes marked with a green outline were indeed found to be modulated after olaparib treatment.

## 3. Results

### 3.1. Olaparib Modulated lncRNA Expression during Tube Formation on Matrigel

PARP inhibitors are currently used in the treatment of tumors displaying homologous recombination deficiencies such as breast and ovarian BRCA1/2-mutant cancers. The in-depth knowledge of the impact of PARP in tumor biology remains a major challenge in order to extend their use to different tumor settings. We have previously shown that PARP inhibitors regulate key aspects of tumor adaptation to their microenvironment with consequences in the control of metastasis.

We previously reported that PARP inhibition could hinder the potential of melanoma cells for VM formation on Matrigel [14]. Moreover, PARP inhibition markedly reduced the expression of VE-cadherin, a well-known marker of VM [13], as well as the phosphorylation of VE-cadherin on Y658, which we reported to be crucial for VM signaling [26]. In this report, we aimed to explore the effect of PARP inhibition on the regulation of non-coding RNA, and specifically of lncRNA, in the context of VM. We have used the cell line Mum2B, a model of highly aggressive UM, capable of undergoing VM both in vitro and in vivo [27]. We seeded Mum2B on Matrigel in order to trigger tube formation. The cell culture was comprised exclusively of UM cells and no endothelial cells, so tube formation is due to the vasculogenic properties of this cell line and not to genuine angiogenesis. Mum2B undergoing tube formation were treated with DMSO or olaparib 5 µM, and total RNA was extracted for RNA sequencing.

In this model, we have found that olaparib significantly modulated the expression of 20 lncRNAs (FDR < 0.05) (Figure 1A), with 11 lncRNAs being upregulated and 9 downregulated (Figure 1B). In order to identify potential *cis*-regulatory effects, we coupled olaparib modulated lncRNAs (*p*-value < 0.05) with immediately upstream and downstream genes. For each *cis*-located gene, we verified if they were modulated by olaparib treatment too (*p*-value < 0.05), and then filtered for significant lncRNA-gene expression interactions (*p*-value < 0.05). A Circos plot was generated to visualize the global alteration in lncRNAs with a predicted *cis*-regulatory role after olaparib treatment, associated with the different chromosomes (Figure 1C). Only genes whose link with a lncRNA is statistically significant are represented in the outer part of the Circos, with their associated lncRNAs represented in the inner part of the Circos.

### 3.2. Olaparib-Modulated lncRNA in the Context of Cancer

For most of the lncRNAs found to be modulated by olaparib, a precise biological function has not been established yet. To find out the overall situation and impact of these lncRNAs in tumor biology, we searched for them in the TCGA PanCancer Atlas cohort in cBioPortal (Figure 2). Uncharacterized genes do not appear in the cBioPortal database, so we performed the search with the 11 lncRNAs which have been officially described so far. In total, 977 out of 10,967 samples (9%) were altered for one or more of these lncRNAs. FLG-AS1 appeared to be altered most often (376 samples, 3% of all samples). On the contrary, LINC01629 was not altered in any sample. Most alterations correspond to amplifications except for RGMB-AS1, which is often deleted in the pan-cancer samples. None of these alterations are identified as driver mutations. The generally large size of lncRNAs makes them prone to accumulate alterations in a genomically unstable context such as cancer, so it should be taken into account that most of the alterations that we found probably fall in the definition of passenger mutations. As shown by the OncoPrint, the information offered in cBioPortal is mostly related to copy number variations (CNV) since this database does not generally annotate sequence information or sequence alterations for non-coding genes or non-coding regions.

When looking separately at the different cancer types (Figure 3 and Appendix A), we found that olaparib-modulated lncRNAs were most frequently altered in sarcoma samples, followed by ovarian epithelial tumors and adrenocortical carcinoma. However, UM samples have almost no alterations in these lncRNAs. Out of 80 samples in the UM cohort, only one was altered, with a deletion in the SNHG4 gene.

Interestingly, in the UM TCGA cohort, SNHG1 expression is negatively correlated with PARP1 and PARP2 (Figure 4), both of which are strongly inhibited by olaparib. On the contrary, SNHG4 and MIR17HG are positively correlated with PARP2. SNHG15 was not correlated with either PARP1 or PARP2. Unfortunately, the data annotated in cBioPortal is mostly focused on coding genes. Information about non-coding genes and lncRNAs, in particular, is limited, and, in this case, there was no available information regarding the gene expression of any other lncRNA in our query.

To further explore this result, we checked if there were any correlations between our lncRNAs and PARP1/2 in SKCM (Appendix A). Again, there was no annotated information on gene expression for most lncRNAs, so only a few lncRNAs can be discussed. SNHG15 and MIR17HG were not correlated to PARP1/2. SNHG1 was positively correlated with PARP2 in this cohort, and SNHG4 was positively correlated with PARP1 in SKCM. This result suggests that there may be a correlation between SNHG1/4 and PARP1/2 mRNA expression levels. Moreover, in the case of SKCM, the correlation is positive, which would support a downregulation of these lncRNAs when PARP activity is inhibited, though it is important to bear in mind that PARP activity is not always correlated with PARP mRNA or protein expression.

Although we cannot compare expression data between different cancer cohorts, there is other information that we can retrieve from the pan-cancer cohort in cBioPortal. We noticed that some of the olaparib-modulated lncRNAs were often altered in the same samples. Indeed, when analyzing mutual exclusivity for alterations in these lncRNAs (Table 1), the results showed that alterations in a number of lncRNAs have a significant co-occurrence. Alterations in SMC5-AS1 and MAMDC2-AS1 have the highest co-occurrence, which can be easily explained by the fact that the loci for these lncRNAs partially overlap. For this reason, a chromosomal rearrangement in that region will very probably affect both genes. SNHG4, RGMB-AS1 and PRR7-AS1 are all located in the long arm of chromosome 5, with SNHG4 and PRR7-AS1 being particularly close, which can explain their higher co-occurrence. FLG-AS1 and LINC01135 are both located in chromosome 1.

We also explored whether alterations in any other genes can co-occur with alterations in olaparib-modulated lncRNAs (Figure 5). Surprisingly, this analysis highlighted a few genes with a critical relevance in cancer. Alterations in TP53 were significantly enriched in samples harboring an amplification in FLG-AS1 (*q*-value = 2.761 × 10^−3^), while this amplification was negatively correlated with alterations in IDH1 (*q*-value = 0.0204) or PTEN (*q*-value = 0.0384) (Figure 5A). Alterations in VHL were significantly enriched in samples with amplifications in SNHG4 (*q*-value = 5.98 × 10^−9^) (Figure 5B) or PRR7-AS1 (*q*-value = 2.94 × 10^−7^) (Figure 5D), and alterations in SPOP were significantly enriched in samples with deep deletion in RGMB-AS1 (*q*-value < 10^−10^) (Figure 5C). It is not surprising that samples with amplifications in SNHG4 and PRR7-AS1 have enrichment for alterations in the same gene since, as stated above, alterations in these two lncRNAs significantly co-occur in pan-cancer samples.

While the illustrated correlations are statistically significant, predicting their biological relevance with bioinformatic tools is not a simple task. As stated above, lncRNAs tend to accumulate passenger alterations due to their large size. Assessing whether the co-occurrence of these alterations with the alterations in TP53, IDH1, PTEN, VHL and SPOP has a biological significance would require further research, which, while not within the scope of the present work, represents an interesting niche for future studies.

As for possible effects on survival, we checked how amplification or deep deletion of olaparib-modulated lncRNAs affected overall survival in pan-cancer patients (Appendix A). Significant differences in survival can only be found for MAMDC2-AS1 (*p* = 0.0365 amplification vs. deletion; *p* = 0.002384 amplification vs. unaltered), SNHG15 (*p* = 2.23 × 10^−5^ amplification vs. unaltered) and LINC01135 (*p* = 0.0138 amplification vs. deletion; *p* = 5.72 × 10^−8^ amplification vs. unaltered). In all cases, amplifications involving those lncRNAs correlated with lower survival. Since olaparib has opposed effects on these lncRNAs, upregulating MAMDC2-AS1, but downregulating SNHG15 and LINC01135, we could say that these results do not show a clear trend towards a benefit from olaparib treatment. However, it is important to point out that in many cases, samples with amplifications in olaparib-modulated lncRNAs have, in fact, whole chromosome amplifications. Therefore, many other genes are altered in these samples, and it is the global effect of these alterations that determines tumor aggressiveness and, eventually, patient survival.

### 3.3. Transcription Factors Upstream Olaparib-Modulated lncRNA

Next, we aimed to find out the possible mechanism by which olaparib could modulate these 20 lncRNAs. Reportedly, PARP1 can alter the DNA-binding activity of basal as well as sequence-specific transcription factors, thereby modulating the transcription of their target genes [28]. Therefore, we hypothesized that the effect of olaparib on lncRNA expression could be due to the ability of PARP to regulate the activity of certain transcription factors. Inhibition by olaparib may interfere with this regulation and so with the expression of target lncRNAs. In an attempt to prove this hypothesis, we studied the promoters of all olaparib-modulated lncRNAs, taking as a promoter the genomic region between −1000 bp and +100 bp from the transcription start site of each lncRNA. Using the JASPAR database, which collects data for transcription factor DNA-binding preferences, we identified all possible transcription factor binding sites with a relative score over 0.95 in each promoter. Results showed that ETS1 had high-score binding sites in the promoters of all olaparib upregulated lncRNAs (Figure 6A), while MZF1, RHOXF1 and NR2C2 had high-score binding sites in the promoters of all olaparib downregulated lncRNAs (Figure 6B).

### 3.4. Predicted lncRNA-Mediated Transcription Perturbations

On the other hand, we thought that olaparib-modulated lncRNAs could regulate the expression of other genes. Furthermore, as reported by other authors, lncRNA can modulate the expression of transcription factors, thereby affecting the expression of all their target genes, which can result in an important shift in the gene expression profile. In this line, we used the database lncMAP to explore whether olaparib-modulated lncRNAs could regulate transcription factor-mediated gene expression (Figure 7). lncMAP uses TCGA data, but it does not collect data for UM. Alternatively, we chose to search in the SKCM cohort. Unfortunately, there was no available data for all olaparib-modulated lncRNAs. However, for the lncRNAs that were annotated in the database, lncMAP showed that each olaparib-modulated lncRNA could further regulate the expression of at least two transcription factors, which can end up affecting a large number of genes. We checked if any of the genes whose expression may potentially be affected by olaparib-modulated lncRNAs were indeed differentially expressed after olaparib treatment in our RNAseq data from Mum2B. This was true for five genes: SCARB1, CYBRD1, FAM49A, NTN4 and NCKAP5L (highlighted in green in Figure 7).

## 4. Discussion

LncRNAs can have a major impact on cell biology since they can modulate several crucial steps of the gene expression process. Several lncRNAs such as HOTAIR or MALAT1 are known to exert an important influence in cancer, but there are many other lncRNAs whose role in cancer remains unexplored. PARP inhibitors are a promising therapy in cancer treatment, having already demonstrated their beneficial effects in the treatment of ovarian cancer patients with specific deficiencies in homologous recombination-mediated DNA repair. The role of PARP in DNA repair has been extensively studied, although PARP can affect nuclear dynamics in many other ways, namely modulating chromatin structure, splicing or transcription [29]. We have previously reported that PARP inhibition can decrease VM formation and VM markers (VE-cadherin and its phosphorylation on Y658) in melanoma [14]. The exact mechanisms responsible for this downregulation remain unknown, and we are currently carrying out several studies concerning the role of PARP in VM. In this study, we explored the potential of PARP in modulating lncRNA expression in cancer and the possible effects of PARP inhibition in modulating the overall gene expression profile in tumor cells. In our VM^+^ UM cell line Mum2B, olaparib significantly modulated the expression of 20 lncRNAs (Figure 1).

FLG-AS1 was the most upregulated lncRNA. There is not much information concerning FLG-AS1 in cancer, though FLG-AS1 was reported to have a higher expression in healthy oral mucosa compared to oral squamous cell carcinoma [30].

The expression of RP3-326I13.1 (also known as PINCR) can increase after DNA damage or upon p53 activation [31]. This lncRNA has been correlated with pro-survival. Its slight upregulation after olaparib treatment may be due to an increase in DNA damage in the absence of PARP activity.

Three of the downregulated genes are small nucleolar RNA (snoRNA) host genes (SNHG), specifically, SNHG1, SNHG15 and SNHG4. SNHGs are regarded generally as oncogenic since most of them have been found to promote common pathways related to aggressiveness and malignancy, such as proliferation, invasion, EMT or apoptosis evasion (reviewed in [32]). SNGH1 has been reported to promote various cancer types, such as hepatocellular carcinoma [33], non-small cell lung cancer [34,35], lung squamous cell carcinoma [36], glioma [37], colorectal carcinoma [38,39], osteosarcoma [40], pancreatic cancer [41] and breast cancer [42], among others. In many of these cancer types, high SNHG1 expression is correlated with poor prognosis and lower progression-free survival. SNHG4 enhanced osteosarcoma [43], cervical cancer [44] and prostate cancer [45]. Finally, SNHG15 promoted renal cell carcinoma [46], gastric [47], pancreatic [48], breast [49], colon [50,51], prostate [52] and non-small cell lung cancer [53], among others. Therefore, the downregulation of these SNHGs by olaparib could have beneficial effects.

RGMB-AS1 can promote lung adenocarcinoma [54], glioma [55] and gastric cancer [56], while MIR17HG can enhance glioma [57], gastric cancer [58] and colorectal cancer [59], among others. These lncRNAs are downregulated by olaparib, too, further suggesting a potential positive effect of olaparib treatment.

Only 11 out of 20 olaparib-modulated lncRNAs were available for query in cBioPortal. 7% of all pan-cancer samples in the TCGA cohort had an alteration in at least one of these 10 lncRNAs (Figure 2). These alterations consist mainly of CNV since sequence mutations in non-coding genes are not generally annotated in this database. However, CNV could still have an important impact on the expression of these lncRNAs and their role in cancer progression.

Interestingly, we found correlations between PARP1 and SNHG1 and SNHG4, and between PARP2 and SNHG1, SNHG4 and MIR17HG (Figure 4 and Appendix A). As explained above, these lncRNAs are all downregulated by olaparib, so their correlation with PARP1/2 should be expected to be in the same direction. A positive correlation, as the one found for SNHG4 and MIR17HG, would be expected in our model since inhibition of PARP1/2 leads to reduced lncRNA expression. Nonetheless, SNHG1 expression is negatively correlated with PARP expression in UM. Nevertheless, it should be taken into account that higher PARP1/2 expression does not necessarily mean higher PARP activity. As a chemical inhibitor, olaparib only affects to PARP1 activity.

Surprisingly, we found that CNV alterations in some olaparib-modulated lncRNAs had a statistically significant correlation with alterations in some key tumor suppressors genes (Figure 5 and Appendix A): samples harboring amplification in FLG-AS1 were frequently mutated in TP53, while there was a negative correlation with mutations in PTEN and IDH1; alterations in SPOP co-occurred with a deep deletion in RGMB-AS1, and alterations in VHL were enriched in samples containing amplification in SNHG4 or in PRR7-AS1. Further research should be carried out in order to elucidate the biological significance of these correlations. As discussed below, these co-occurrences might be passenger effects due to the higher incidence of these alterations in certain tumor types.

TP53 codes for the well-known tumor suppressor p53. TP53 is the most frequently mutated gene in cancer, reportedly mutated in almost 50% of human tumors, which is associated with poor patient prognosis. Wildtype p53 can arrest cell cycle progression after DNA damage, hindering the accumulation of potentially oncogenic mutations in the cell. Malfunction of p53 can lead to evasion of tumor cell death, which in turn allows tumor growth [60,61]. TP53 is altered in over half of the samples containing FLG-AS1 amplification (192 out of 376 samples) (Appendix A), having a significant co-occurrence in the pan-cancer cohort. As expected for this gene, most TP53 alterations in those 192 samples are classed as putative drivers, most of the missense mutations, but also truncating mutations and a few in-frame mutations. Samples harboring alterations in both genes are mostly from non-small cell lung cancer and invasive breast carcinoma.

The phosphatase and tensin homolog (PTEN) is a potent tumor suppressor too, and also one of the most frequently mutated genes in human cancer. PTEN is involved in a number of cellular processes, such as proliferation, cell survival and maintenance of genomic stability. Even slight decreases in PTEN activity have been reported to promote tumor growth [62]. In the TCGA pan-cancer cohort, PTEN alterations occur most frequently in uterine corpus endometrial carcinoma (UCEC) (66.54% of all UCEC samples; data not shown), while FLG-AS1 amplification is most prevalent in hepatocellular carcinoma, lung adenocarcinoma and breast invasive carcinoma (Appendix A), with only two UCEC samples harboring an amplification in this lncRNA. This unequal distribution across cancer types explains why FLG-AS1 amplification and PTEN alterations appear as mutually exclusive in the pan-cancer context.

Isocitrate dehydrogenase 1 and 2 (IDH1/2) mutations are also frequent in several types of cancer, especially in a number of brain tumors. In fact, IDH1/2 mutations are indicators of good prognosis in astrocytomas and glioblastomas since these mutations can lead to improved responses to irradiation and chemotherapy [63,64]. IDH1 alterations were mostly found in the lower grade glioma (LGG) TCGA cohort in cBioPortal, with 78% of all LGG samples having alterations in this gene (data not shown). However, as stated above, FLG-AS1 amplification is most common in other cancer types, and was only present in two LGG samples, hence the mutual exclusivity between alterations in these genes in the pan-cancer context.

The nuclear speckle-type pox virus and zinc finger protein (SPOP) is a substrate-recognition subunit of cullin-RING E3 ligases [65]. The co-occurrence between SPOP mutations and RGMB-AS1 deletion happened exclusively in prostate adenocarcinoma (PRAD). In fact, RGMB-AS1 was most frequently altered in this cancer type (Appendix A), and SPOP was altered in 12% of samples in the PRAD TCGA cohort (Appendix A), though according to some publications [66], SPOP could be altered in up to 15% of prostate tumors. SPOP alterations and RGMB-AS1 deletions significantly co-occurred in the PRAD cohort, with 20 out of 489 samples containing alterations in both genes (Appendix A). RGMB-AS1 was deleted in all these 20 samples, while SPOP had suffered some kind of missense mutation, affecting the residues Y87, F102, F125, K129, W131 or F133. Most of these residues are part of SPOP substrate-binding cleft [65], and all mutations were classed as putative drivers of cancer. Deletions in the chromodomain helicase DNA binding protein 1(CHD1) have also been associated with SPOP mutations [66]. A query for SPOP, CHD1 and deletions in RGMB-AS1 in the PRAD cohort (Appendix A) confirmed a co-occurrence between alterations in all three genes. In fact, CHD1 and RGMB-AS1 are located in the same cytogenetic band (5q15) in the genome; hence a deletion in RGMB-AS1 always involves a deletion in CHD1 too. To our knowledge, there are no available publications analyzing the role of RGMB-AS1 in prostate cancer, though it would be of great interest to find out if the pro-tumoral effect of deletions in the 5q15 region is due exclusively to the loss of CHD1 or if RGMB-AS1 could be playing an important role as well. This is of particular interest since SPOP mutation in PRAD has been reported to impair homologous recombination-mediated DNA repair, sensitizing to PARP inhibitors, and CHD1 deletions have been associated with increased nonhomologous end joining (NHEJ) DNA-repair [67], which could lead to sensitivity to PARP inhibitors as well.

The von Hippel–Lindau (VHL) tumor suppressor gene is well known for causing VHL disease, which involves predisposition to several cancer types, namely clear cell renal cell carcinoma (ccRCC). Loss of VHL interferes with the degradation of hypoxia-inducible factors HIF1α and HIF2α and therefore promotes the expression of hypoxic genes, which is directly associated with tumorigenesis [68]. VHL alterations were present in 16 samples with amplification in SNHG4 (Appendix A), all of them ccRCC except a PRAD sample. As for samples with PRR7-AS1 amplifications, 20 of them also had alterations in VHL (Appendix A), 16 of those being ccRCC too. In fact, ccRCC was the cancer type where both SNHG4 and PRR7-AS1 were most frequently altered (Appendix A). VHL alterations in these samples were mainly truncating mutations classed as putative drivers. To our knowledge, there are no available publications researching the role of SNHG4 or PRR7-AS1 in ccRCC.

In an attempt to find out how olaparib affected lncRNA expression, we analyzed the promoters of olaparib-modulated lncRNAs for transcription factor binding sites. All olaparib-upregulated lncRNAs had binding sites for ETS1, while all olaparib-downregulated lncRNAs had binding sites for MZF1, RHOXF1 and NR2C2. We hypothesized that the activity of these transcription factors could be directly modulated by PARP, which would explain the effect of olaparib in regulating the expression of their target lncRNA. Indeed, we found several reports correlating PARP1 with some of these transcription factors, as detailed below.

ETS1 is the founding member of the ETS domain family of transcription factors, consisting of 28 ETS genes in humans. The specific function of ETS1 highly depends on the tissue and the cellular context. ETS1 has been associated with cancer progression in a broad range of cancer types, where ETS1 expression is often correlated with undifferentiated status, higher invasion and angiogenesis, increased metastasis and drug resistance [69]. Interestingly, PARP1 has been repeatedly shown to be a target gene of ETS1 transcriptional activity [70,71] in Ewing sarcoma. Moreover, PARP1 has been reported to interact with ETS1 [72,73], and PARP inhibition promoted the nuclear accumulation of Ets-1 and increased its transcriptional activity. This supports our results, where lncRNAs containing ETS1 binding sites in their promoters were upregulated upon PARP inhibition. Moreover, ETS1 can play a crucial role in tumor vasculature by modulating endothelial gene expression [69], so it raises the question of whether olaparib could somehow regulate VM in our model via promoting ETS1 activity.

Myeloid zinc finger 1 (MZF1) is a member of the SCAN-zinc finger family of transcription factors. Amplifications in the MZF1 gene, as well as high MZF1 expression, are frequent in cancer, as shown by TCGA data, and MZF1 has been associated with cancer progression in a variety of solid tumors [74]. Surprisingly, Fang et al. have previously reported that PARP1 can upregulate MZF1 expression in an E2F1-mediated manner in neuroblastoma [75]. In fact, this upregulation was promoted by a lncRNA, MZF1-AS1, which bound PARP1 to increase its interaction with E2F1 and, therefore, the upregulation of MZF1. Fang et al. showed that the effect of PARP1 in MZF1 expression was independent of PARylation, in line with the first report of an interaction between PARP1 and E2F1 [76], although other authors have shown that PARP1 can indeed PARylate E2F1 [77]. In our study, the inhibition of PARP activity with olaparib downregulated lncRNAs containing MZF1 binding sites in their promoters. This supports a PARylation-dependent interaction between PARP1 and MZF1 activity where PARP inhibition could reduce MZF1-dependent gene expression.

RHOXF1 (reproductive homeobox on X-chromosome F1) is predominantly expressed in the testis in healthy adults, where it is essential for male fertility [78], but its expression was also found in a number of tumor cell lines [79]. Our results suggest that olaparib could decrease RHOXF1 activity, which may implicate reduced stemness, though in this case, there are no previous reports correlating PARP and RHOXF1.

The nuclear receptor superfamily 2 group C member 2 (NR2C2), also known as testicular nuclear receptor 4 (TR4), has been associated with tumorigenesis in Cushing’s syndrome [80]. It has also been reported as a relevant factor in prostate cancer, although it can have opposing effects depending on the stage and on PPARγ status [81]. Moreover, NR2C2 reportedly promoted VM in ccRCC [82]. Our study suggests that olaparib may decrease NR2C2-dependent lncRNA expression. It would be interesting to look more closely at the possible modulation of NR2C2 activity by olaparib and its potential implications in VM.

As stated above, one of the many ways in which lncRNAs can modulate gene expression is via specific interactions with transcription factors that can modify their transcriptional activity [83]. In this context, a change in the expression of one lncRNA can further affect the expression of many other genes. In our study, the expression of twenty lncRNAs was significantly changed after olaparib treatment. In an attempt to estimate the potential effect that this could have on global gene expression, we used the database lncMAP [25] and searched lncRNA- transcription factor- gene triplets for each olaparib-modulated lncRNA (Figure 7). lncMAP does not contain any data for UM, though the authors reported that lncRNA-mediated gene expression perturbations have a similar pattern in cancer types with similar origins, so we performed our search in SKCM. Moreover, though the authors showed that up to 17% of lncRNA modulation patterns were highly cancer-specific, they also found some pan-cancer lncRNA modulators, where a given lncRNA took part in the same lncRNA- transcription factor- gene triplet in over 15 different cancer types. In line with this, we found five genes (SCARB1, CYBRD1, FAM49A, NTN4 and NCKAP5L) predicted to be affected by lncRNA changes in SKCM, which indeed suffered a significant change in response to olaparib in UM. Therefore, though the SKCM lncRNA-mediated expression network may not be the ideal approach to find out the exact gene expression changes in our UM model, it can still give us a general idea of how much the gene expression profile can change in response to olaparib-mediated lncRNA expression changes in any given cancer type.

In conclusion, using integration analysis of lncRNA from cells undergoing VM after olaparib treatment, potential candidate target genes with strong clinical significance were identified, providing novel insights into the molecular pathological mechanisms underlying VM. However, the findings of the present study were limited by the relatively small sample size of UM in public databases. Future studies should involve the cooperation of basic and clinical research to further verify the findings of the present study and should investigate the mechanisms underlying the development of VM in UM, potentially leading to the identification of novel diagnostic and therapeutic biomarkers.

## Figures and Tables

**Figure 1 cells-09-02690-f001:**
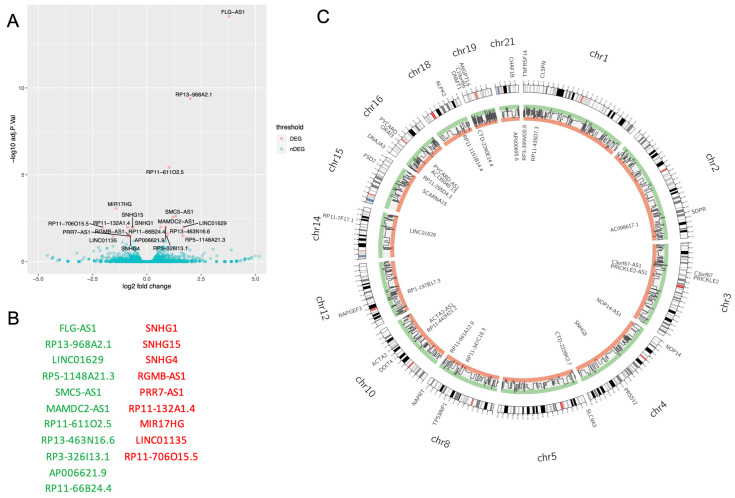
Olaparib modulated the expression of 20 long non-coding RNAs (lncRNAs). (**A**) volcano plot displaying lncRNAs whose expression is changed by olaparib treatment. Shown in red are 20 lncRNAs whose expression changes significantly (FDR < 0.05). (**B**) lncRNAs whose expression changes significantly after olaparib treatment, ranked from highest to lowest logFC. logFC > 0 are shown in green; logFC < 0 are shown in red. (**C**) lncRNAs which might regulate gene expression in *cis* following olaparib treatment. The inner side of the Circos shows lncRNA, while the outer side shows genes whose expression might be *cis*-regulated by each lncRNA. The Circos shows a total of 24 pairs each made up of one olaparib-modulated lncRNA (*p*-value < 0.05) and one olaparib-modulated gene (*p*-value < 0.05) whose relationship is statistically significant (*p*-value < 0.05).

**Figure 2 cells-09-02690-f002:**
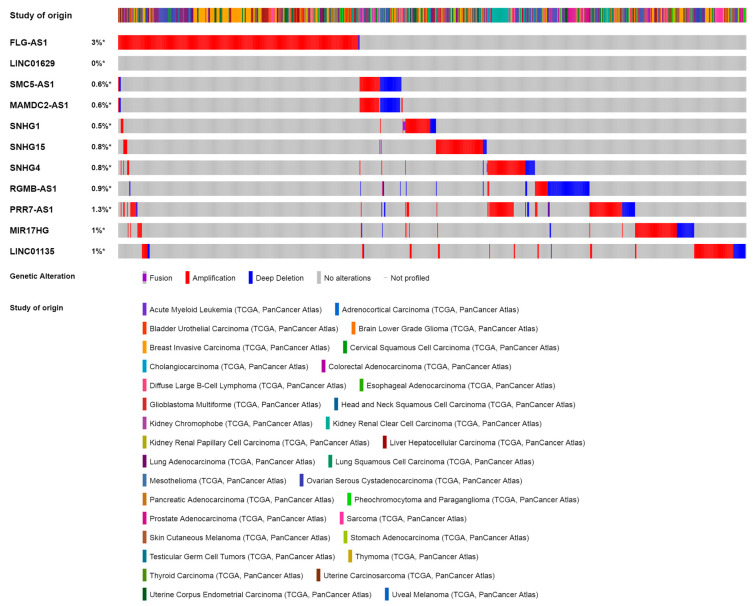
Olaparib-modulated lncRNAs in pan-cancer samples. OncoPrint for olaparib-modulated lncRNAs in the PanCancer Atlas cohort, obtained from cBioPortal. 977 out of 10,967 (9%) cancer samples are altered for one or more of these lncRNAs. * *p*-value < 0.05.

**Figure 3 cells-09-02690-f003:**
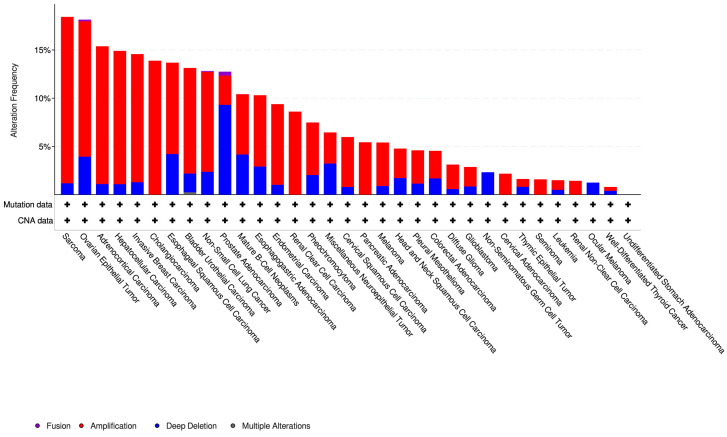
Alteration of olaparib-modulated lncRNAs across cancer types. Overall alteration frequency of olaparib-modulated lncRNAs across different cancer types in the PanCancer Atlas cohort, obtained from cBioPortal.

**Figure 4 cells-09-02690-f004:**
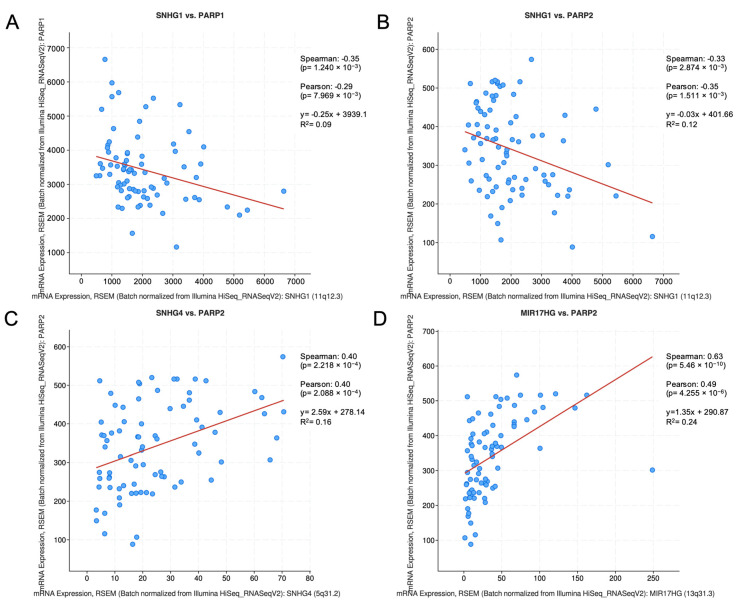
Co-expression between olaparib-targeted poly-(ADP-ribose) polymerases (PARP) and olaparib-modulated lncRNAs. Co-expression between PARP1/2 and olaparib-modulated lncRNAs in the uveal melanoma cohort from The Cancer Genome Atlas (TCGA), obtained from cBioPortal. Graphs show a significant correlation (*q*-value < 0.05) between (**A**) mRNA expression of SNHG1 and mRNA expression of PARP1; (**B**) mRNA expression of SNHG1 and mRNA expression of PARP2; (**C**) mRNA expression of SNHG4 and mRNA expression of PARP2; (**D**) mRNA expression of MIR17HG and mRNA expression of PARP2.

**Figure 5 cells-09-02690-f005:**
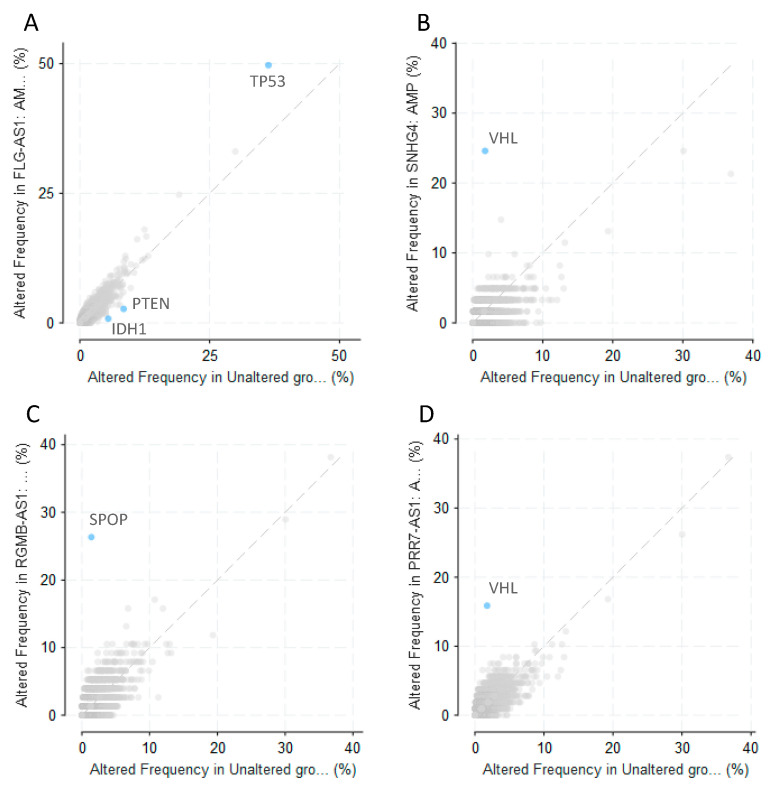
Co-alteration between olaparib-modulated lncRNAs and other genes in cancer. Correlation between alterations in olaparib-modulated lncRNAs and alterations in other genes in samples from the PanCancer Atlas cohort, obtained from cBioPortal. (**A**) Enrichment of amplification in FLG-AS1 in samples with other alterations. Amplification in FLG-AS1 is positively correlated with alterations in TP53 (*q*-value = 2.761 × 10^−3^) and negatively correlated with alterations in IDH1 (*q*-value = 0.0204) and in PTEN (*q*-value = 0.0384); (**B**) enrichment of amplification in SHNG4 in samples with other alterations. Amplification in SNHG4 is positively correlated with alterations in VHL (*q*-value = 5.98 × 10^−9^); (**C**) enrichment of deep deletion in RGMB-AS1 in samples with other alterations. Deep deletion in RGMB-AS1 is positively correlated with alterations in SPOP (*q*-value < 10^−10^) (**D**); enrichment of amplification in PRR7-AS1 in samples with other alterations. Amplification in PRR7-AS1 is positively correlated with alterations in VHL (*q*-value = 2.94 × 10^−7^).

**Figure 6 cells-09-02690-f006:**
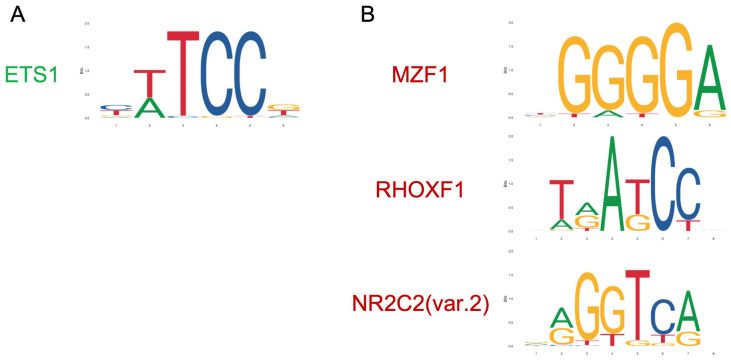
Transcription factors predicted to control the expression of lncRNA in response to olaparib. Transcription factors with potential binding sites in the promoter region (−1000 bp to +100 bp from the transcription start site) of (**A**) olaparib-upregulated lncRNAs and (**B**) olaparib-downregulated lncRNAs, as predicted by JASPAR.

**Figure 7 cells-09-02690-f007:**
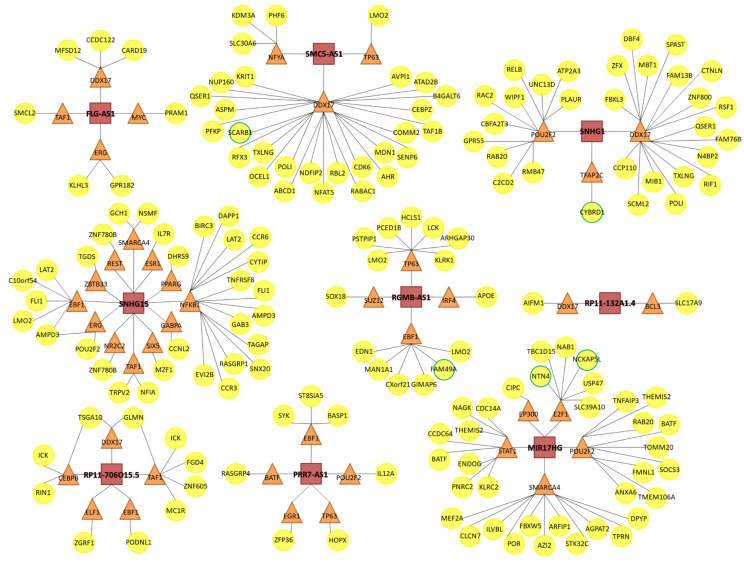
Predicted lncRNA-mediated transcription perturbations in response to olaparib. Diagrams illustrating data from the lncMAP database with regard to olaparib-modulated lncRNA-transcription factor-gene triplets in the skin cutaneous melanoma (SCKM) TCGA cohort. Olaparib-modulated lncRNAs are shown as red squares. Transcription factors are shown as orange triangles. Target genes are shown as yellow circles. Genes highlighted in green suffered a significant change in gene expression after olaparib treatment in our model.

**Table 1 cells-09-02690-t001:** Mutual exclusivity of alterations in olaparib-modulated lncRNAs in the PanCancer Atlas cohort from cBioPortal, showing how alterations in some of these lncRNAs tend to co-occur.

A	B	Neither	A Not B	B Not A	Both	Log2 Odds Ratio	*p*-Value	*q*-Value	Tendency
SMC5-AS1	MAMDC2-AS1	10,119	3	2	65	>3	<0.001	<0.001	Co-occurrence
SNHG4	PRR7-AS1	10,018	34	93	44	>3	<0.001	<0.001	Co-occurrence
RGMB-AS1	PRR7-AS1	9962	90	128	9	2.96	<0.001	<0.001	Co-occurrence
SNHG4	RGMB-AS1	10,019	71	92	7	>3	<0.001	<0.001	Co-occurrence
SNHG1	PRR7-AS1	10,002	50	131	6	>3	<0.001	0.001	Co-occurrence
FLG-AS1	PRR7-AS1	9693	359	122	15	1.731	<0.001	0.001	Co-occurrence
SMC5-AS1	RGMB-AS1	10,027	63	94	5	>3	<0.001	0.004	Co-occurrence
FLG-AS1	LINC01135	9720	362	95	12	1.762	<0.001	0.004	Co-occurrence
PRR7-AS1	MIR17HG	9950	131	102	6	2.16	0.003	0.02	Co-occurrence
MAMDC2-AS1	RGMB-AS1	10,027	63	95	4	2.744	0.004	0.022	Co-occurrence

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
