# Peer review of "The PARP Inhibitor Olaparib Modulates the Transcriptional Regulatory Networks of Long Non-Coding RNAs during Vasculogenic Mimicry"

_cells, 2020, doi:10.3390/cells9122690_

Round 1

Reviewer 1 Report

This paper is about the role of Olaparib to modulate the transcriptional regulatory networks of long noncoding RNAs during vasculogenic mimicry. The readers expect to read something about the netwrok sciences, network construction and analysis, three layer regulatory network and etc. But honestly, I could not find anything in abstract, introduction and method about the network. Just it appeared in title and Figure 7 without any description that how the athours created and interpreted it. These are my main concerns that I cannot recommend this MS for publication in the current form, otherwise it is well written. I just mentioned some of my concerns as follows:

1-The abstract is not informative and to some extent vague. I recommend the authors to rewrite it.

2- Actually, the Intoduction portion is a summary of the rols of lncRNAs that is not good enough for this manuscript. The authors should do a survey on the rols of lncRNA in cancers with the focuse on VM. The authors did not mention VM and angiogenesis in the MS at all. It should be totally rewritten.

3- How the authors distinguish VM and angiogenesis in the cell culture. The authors just mentioned that "uveal melanoma cells capable of undergoing pseudoangiogenesis", but they did not prove it.

4- I can not follow the results. I think it should be catagorized as method section. Which part of the results belongs to method subsections.

5- How did you creat the network (Figure. 7). The athours did not mention in method section at all.

Author Response

Referee 1

Responses (R):

We would like to sincerely thank all the referees for their in-depth review of our manuscript which has helped us to improve the current version.

1-The abstract is not informative and to some extent vague. I recommend the authors to rewrite it.

R. We have re-organised and completed the information of the abstract to distinguish the alteration induced by olaparib in two layers of transcriptional regulation,

“diverse regulatory networks have been found to be altered by olaparib: Copy number variation (CNV) alterations in some olaparib-modulated lncRNAs had a significant correlation with alterations in some key tumor suppressor genes, including TP53, PTEN, SPOP, VHL and IDH1. Importantly, olaparib also regulated the expression of a set of lncRNAs regulated by common transcription factors: ETS1 had high-score binding sites in the promoters of all olaparib upregulated lncRNAs while MZF1, RHOXF1 and NR2C2 had high-score binding sites in the promoters of all olaparib downregulated lncRNAs.”

2- Actually, the Intoduction portion is a summary of the rols of lncRNAs that is not good enough for this manuscript. The authors should do a survey on the rols of lncRNA in cancers with the focuse on VM. The authors did not mention VM and angiogenesis in the MS at all. It should be totally rewritten.

R. As we did in the abstract, we have re-organised the introduction and included new information about tumor angiogenesis, vasculogenic mimicry and the role of lncRNAs in their regulation.

3- How the authors distinguish VM and angiogenesis in the cell culture. The authors just mentioned that "uveal melanoma cells capable of undergoing pseudoangiogenesis", but they did not prove it.

R. In the result section we have included the following explanation:

“We seeded Mum2B on matrigel in order to trigger tube formation (27). The cell culture was comprised exclusively of uveal melanoma cells and no endothelial cells, so tube formation is due to the vasculogenic properties of this cell line, and not to genuine angiogenesis. Mum2B undergoing tube formation were treated with DMSO or olaparib 5 µM, and total RNA was extracted for RNA sequencing”.

We are also attaching a figure (not to be published) for the referee indicating that Olaparib decreased covered area in an in vitro VM assay. An on-going project of our group is investigating different aspects of the role of PARP1 in VM development and in the current manuscript we have focalized in the alteration of lncRNAs by olaparib.

4- I cannot follow the results. I think it should be catagorized as method section. Which part of the results belongs to method subsections.

R. With the aim to facilitate an improved comprehension of the results we have sub-divided the results section with similar headings to the methods section.

5- How did you creat the network (Figure. 7). The athours did not mention in method section at all.

R. At this respect we have now included an explanation in the method sub-section 2.8 as follows:

“We searched each olaparib-modulated lncRNA in the skin cutaneous melanoma cohort. Unfortunately, not all lncRNA were available in this database. For those that were, lncMAP provided a list of transcription factors and downstream genes that were predicted to be regulated by each lncRNA (Supplemental Material, excel document), ranking lncRNA- transcription factor- gene triplets by FDR. Using this data, we made a graphic representation (Figure 7) of the 30 most significant triplets predicted for each lncRNA, connecting every lncRNA (that we represented as a red square) to each transcription factor (orange triangle) that it was predicted to modulate, and further connecting every transcription factor to each gene (yellow circle) whose expression was predicted to change in response to the lncRNA-mediated transcription factor modulation. “

We have also attached as supplemental material an excel document with the results represented in the figure 7, as obtained from the mentioned database.

Reviewer 2 Report

In the manuscript entitled “The PARP inhibitor Olaparib modulates the transcriptional regulatory networks of long noncoding RNAs during vasculogenic mimicry”, Fernandez-Cortes et al. studied the effects of PARP inhibitor Olaparib on metastatic tumors vasculogenic mimicry, especially in association with lncRNAs. However, the study is basically the observation. Mechanistic correlation is completely lacked. It is not shown how PARP inhibitor affects those expressions. It is not shown how those expression alterations really affect. I do not think this manuscript is enough informative for publication.

Major problems:

  1. PARP1/2 mediate multiple cellular pathways, including multiple DNA repair pathways, epigenetic regulation, and transcription. Therefore, a number of cellular pathways are modulated. It is not impressive if the mechanisms for those alterations are clarified.

  1. It is also not shown if there are any phenotype alterations through the pathways of PARP inhibition and lncRNA expression alterations.

  1. Although the authors stated that “Copy number variation (CNV) alterations in some olaparib-modulated lncRNAs had a significant correlation with alterations in some key tumor suppressor genes, including TP53, PTEN, SPOP, VHL and IDH1”, it is not shown how those correlation is important and responsible for the resulting cellular phenotype. Without those studies, we do not really know how those correlations what the authors found are important or just passenger effects.

  1. Although the authors claimed “Overall, we propose that olaparib treatment may trigger a major transcriptome shift mediated by the regulation of the lncRNA landscape”, I have to strongly disagree to this conclusion, because they have simply observed the association with an experiment using a cell type. The authors at least have to show with multiple systems using multiple cell types.

Minor points:

  1. Letters written in Figure 1C is too faint to see. What are those?
  2. In Figure 4, although the authors claimed some correlations between PARP and lncRNAs, it is not shown how is in others.

Author Response

We would like to sincerely thank all the referees for their in-depth review of our manuscript which has helped us to improve the current version.

Responses to Referee 2:

Major problems:

  1. PARP1/2 mediate multiple cellular pathways, including multiple DNA repair pathways, epigenetic regulation, and transcription. Therefore, a number of cellular pathways are modulated. It is not impressive if the mechanisms for those alterations are clarified.                                                                     R. We appreciate this comment and we have included additional information to clarify the possible mechanism by which PARP1 may affect lncRNAs regulation. The implication of different PARPs (particularly PARP-1) in the modulation of transcription at the level of modification of chromatin structure, modulation of insulator function and direct alteration in the transcriptional machinery, has been extensively reported (Recently reviewed by Gupte et al., Genes and Dev. doi: 10.1101/gad.291518.116.) and this effect mediated by PARP-1 is independent of its action in DNA repair pathways. In this respect, in the discussion we had already referred to the reported interaction between PARP-1 and two of the transcription factors identified by JASPAR: ETS1, which has binding sites in the promoters of all olaparib-upregulated lncRNAs, and MZF1, which has binding sites in the promoters of all of the olaparib-downregulated lncRNAs (MZF1). Como se detalla
  1. It is also not shown if there are any phenotype alterations through the pathways of PARP inhibition and lncRNA expression alterations.                     R. Our group has previously shown that PARP inhibition decreased VM in vitro and down-regulated the expression and phosphorylation of VE-cadherin in melanoma cells, which is a key marker involved in VM (Rodriguez-Lara, Peralta-Leal et al , PLOS Genetics 2013; Delgado-Bellido et al., Cell Death Diff 2018). Here is the text that we have included in the manuscript to answer to your request.

“We previously reported that PARP inhibition can hinder the potential of melanoma cells for vasculogenic mimicry (VM) formation on matrigel (8). Moreover, PARP inhibition markedly reduced the expression of VE-cadherin, a well-known marker of VM (20), as well as the phosphorylation of VE-cadherin on Y658, which we reported to be crucial for VM signaling (21). In this report we aimed to explore the effect of PARP inhibition on the regulation of non-coding RNA, and specifically of lncRNA, in the context of VM..”

We are also attaching a figure (not to be published) for the referee indicating that Olaparib decreased covered area in an in vitro VM assay. An on-going project of our group is investigating different aspects of the role of PARP1 in VM development and in the current manuscript we have focalized in the alteration of lncRNAs by olaparib.

  1. Although the authors stated that “Copy number variation (CNV) alterations in some olaparib-modulated lncRNAs had a significant correlation with alterations in some key tumor suppressor genes, including TP53, PTEN, SPOP, VHL and IDH1”, it is not shown how those correlation is important and responsible for the resulting cellular phenotype. Without those studies, we do not really know how those correlations what the authors found are important or just passenger effects.                                                                                                      R. We appreciate this criticism. We have to keep in mind that we relay in experimental data issued from our RNA-seq and the bioinformatic analysis results in a statistically significant correlation between some olaparib modulated lncRNAs, as represented in figure 5. We consider relevant this information given the extended use of olaparib as anti-tumor agent and the key function of the tumor suppressor genes identified in this analysis. As the referee states, further confirmation will be needed to actually show how relevant these correlations are. However, performing this kind of experiment will take a long time, and will be from the main objective of the current study that gives an integrative view of the alterations derived form the use of olaparib in VM.

  1. Although the authors claimed “Overall, we propose that olaparib treatment may trigger a major transcriptome shift mediated by the regulation of the lncRNA landscape”, I have to strongly disagree to this conclusion, because they have simply observed the association with an experiment using a cell type. The authors at least have to show with multiple systems using multiple cell types.                                                                                                       R. We again appreciate this suggestion. This specific point will be ideally a significant add to our current study but it is not realistic to perform a RNAseq of a different cell line with vasculogenic mimicry properties, which would be completely out of a reasonable period of time to conclude our study.

Minor points:

  1. Letters written in Figure 1C is too faint to see. What are those                       R. We include a new figure 1 with largers letters for the circos plot
  1. In Figure 4, although the authors claimed some correlations between PARP and lncRNAs, it is not shown how is in others.                                                     R. As we have already specified in the text we have analysed the correlation of the lncRNAs that were availables in the databases.

Reviewer 3 Report

The author by Mónica Fernández-Cortés et al titled The PARP inhibitor Olaparib modulates the transcriptional regulatory networks of long noncoding RNAs during vasculogenic mimicry" study was designed well.  They analysed lncRNAs vasculogenic mimicry (VM) and  the whole transcriptome profiling of human uveal melanoma cells significantly modulated and their finding shows 11 lncRNAs  was upregulated and 9 IncRNAs was downregulated. This study concluded that olaparib modulated lncRNA expression during VM affecting angiogenesis related genes.   I have done the plagiarism and there are no major issues. I have few minor comments.   Please find the below minor comments.
  1.  The author needs to mention references few places.
  2.  Each figure should have the headings and it should be clearly mentioned in the results section also.

Author Response

We would like to sincerely thank all the referees for their in-depth review of our manuscript which has helped us to improve the current version.

Here are the responses to the referee (R):

  1.  The autor needs to mention references few places.

R. We have carefully ckeck te text and we have included all the corresponding references

  1.  Each figure should have the headings and it should be clearly mentioned in the results section also.

R. In the new version we have included a heading for each figure

Round 2

Reviewer 1 Report

The authors addressed all my concerns and I think the paper is in a good shape for publication in cells. Please carefully proof-read spell check to eliminate grammatical errors before publication.

Line 40 -> LncRNAs play

Line 63 -> LncRNAs have been proposed

Line 65 -> lncRNA HIF2PUT acts

Line 72 -> LINC00312 induces

Line 73 -> (11).

Line 73 -> SNHG20 promotes

Line 74 -> which results

Line 76 -> VM (13)

Line 81 -> vasculogenic mimicry -> VM

Line 151 -> all lncRNAs

Line 188 -> vasculogenic mimicry -> VM

Author Response

We thank you so much for your helpful review. In the current version we have amended all the mistakes you found in opur previous version.

Reviewer 2 Report

In the manuscript entitled “The PARP inhibitor Olaparib modulates the transcriptional regulatory networks of long non-coding RNAs during vasculogenic mimicry”, Fernández-Cortés et al. studied the effects of PARP inhibitor Olaparib on metastatic tumors vasculogenic mimicry, especially in association with lncRNAs. This is the revised manuscript, in which I found some improvement. However, the study is basically the observation and no mechanistic correlations. Seems to me, the authors are just looking at multiple passenger effects after Olaparib treatment. In fact, none of the data really supports the authors’ original work motivation, i.e., the identification of the antimetastatic target. Therefore, I still do not think this manuscript is enough informative for publication.

Major problems:

1: Although the original aim of this study is to identify the antimetastatic targets by Olaparib treatment, the study is not designed to identify the targets.

2: Related to the first problem, although the authors observed some lncRNAs that are altered in the presence of Olaparib, the study is not even designed to address whether such lncRNA level alterations are associated with antimetastatic phenotype expression. As it is well known, the critical driving regulation is usually expressed together with a number of passenger alterations. In fact, passenger alterations are much more. Therefore, the authors major observation by itself is not really addressing the authors’ original point.

If the authors are really thinking such lncRNA level alterations are associated with antimetastatic phenotype expression, this issue must be addressed.

3: Although the authors claim the association between PARP inhibition and lncRNA expressions, the mechanistic link is again lucked.

4: It is also not clear why the authors focus only on lncRNAs. True, some lncRNAs might be associated with antimetastatic effect; however, there are many other effects as well.

5: In addition, the authors did not really address the previous criticisms, as well.

Author Response

We thank the referee for the effort to evaluate carefully our manuscript.

Please find below a point-by-point response to all the concerns raised for our previous version:

In the manuscript entitled “The PARP inhibitor Olaparib modulates the transcriptional regulatory networks of long non-coding RNAs during vasculogenic mimicry”, Fernández-Cortés et al. studied the effects of PARP inhibitor Olaparib on metastatic tumors vasculogenic mimicry, especially in association with lncRNAs. This is the revised manuscript, in which I found some improvement. However, the study is basically the observation and no mechanistic correlations. Seems to me, the authors are just looking at multiple passenger effects after Olaparib treatment. In fact, none of the data really supports the authors’ original work motivation, i.e., the identification of the antimetastatic target. Therefore, I still do not think this manuscript is enough informative for publication.

Major problems:

1: Although the original aim of this study is to identify the antimetastatic targets by Olaparib treatment, the study is not designed to identify the targets.

As we stated in the abstract, we chose to search for potential lncRNAs modulated by olaparib and potentially implicated in metastasis, using a highly metastatic cell line, MUM2B, cultured in matrigel to develop vasculogenic mimicry to simulate the conditions during tumor development/metastasis. To achieve this aim we used an unbiased method; by doing so, we were able to identify lncRNAs that have been already implicated in metastasis according to the previous literature, such as SNHG1, SNHG15, SNHG4, RGMB-AS1 and MIR17HG, whose relevance in tumor development is developed in the discussion section (lines 348-399; references 34-60).

Nevertheless, we have taken this comment into account and changed the wording of the abstract as follows:

“To evaluate the impact of olaparib treatment in the context of non-coding RNA we have analyzed the expression of lncRNA after performing an unbiased whole-transcriptome profiling of human uveal melanoma cells cultured to form VM.”

2: Related to the first problem, although the authors observed some lncRNAs that are altered in the presence of Olaparib, the study is not even designed to address whether such lncRNA level alterations are associated with anti-metastatic phenotype expression. As it is well known, the critical driving regulation is usually expressed together with a number of passenger alterations. In fact, passenger alterations are much more. Therefore, the authors major observation by itself is not really addressing the authors’ original point.

If the authors are really thinking such lncRNA level alterations are associated with antimetastatic phenotype expression, this issue must be addressed.

As expressed in the previous response and in the revised version already submitted, we have identified lncRNAs not only associated with metastasis but that have been demonstrated to be driver alterations.

We quote the text of the discussion below:

“Three of the downregulated genes are small nucleolar RNA (snoRNA) host genes (SNHG), specifically, SNHG1, SNHG15 and SNHG4. SNHGs are regarded generally as oncogenic, since most of them have been found to promote common pathways related with aggressiveness and malignancy, such as proliferation, invasion, EMT or apoptosis evasion (reviewed in (33)). SNGH1 has been reported to promote various cancer types, such as hepatocellular carcinoma (34), non-small cell lung cancer (35,36), lung squamous cell carcinoma (37), glioma (38), colorectal carcinoma (39,40), osteosarcoma (41), pancreatic cancer (42) and breast cancer (43), among others. In many of these cancer types, high SNHG1 expression is correlated with poor prognosis and lower progression free survival. SNHG4 enhanced osteosarcoma (44), cervical cancer (45) and prostate cancer (46). Finally, SNHG15 promoted renal cell carcinoma (47), gastric (48), pancreatic (49), breast (50), colon (51,52), prostate(53) and non-small cell lung cancer (54), among others. Therefore, the downregulation of these SNHGs by olaparib could have beneficial effects.

RGMB-AS1 can promote lung adenocarcinoma (55), glioma (56) and gastric cancer (57), while MIR17HG can enhance glioma (58), gastric cancer (59) and colorectal cancer (60), among others. These lncRNAs are downregulated by olaparib too, further suggesting a potential positive effect of olaparib treatment.”

3: Although the authors claim the association between PARP inhibition and lncRNA expressions, the mechanistic link is again lucked.

Concerning this point, in the response to the referee 2 included in our revised version we already answered as follows:

“We appreciate this comment and we have included additional information to clarify the possible mechanism by which PARP1 may affect lncRNAs regulation. The implication of different PARPs (particularly PARP-1) in the modulation of transcription at the level of modification of chromatin structure, modulation of insulator function and direct alteration in the transcriptional machinery, has been extensively reported (Recently reviewed by Gupte et al., Genes and Dev. doi: 10.1101/gad.291518.116.) and this effect mediated by PARP-1 is independent of its action in DNA repair pathways. In this respect, in the discussion we had already referred to the reported interaction between PARP-1 and two of the transcription factors identified by JASPAR: ETS1, which has binding sites in the promoters of all olaparib-upregulated lncRNAs, and MZF1, which has binding sites in the promoters of all of the olaparib-downregulated lncRNAs (MZF1)”.

References 73-74 concern the interaction between PARP1 and ETS1. References 75-78 concern the interaction between PARP1 and MZF1. In both cases, PARP inhibition could modulate the expression of target genes. We propose that the lncRNA discussed in our report are indeed downstream targets of ETS1 and MZF1, based on the in silico analysis of lncRNA promoters with JASPAR, and that would be why there is a change in lncRNA expression in response to PARP inhibition.

4: It is also not clear why the authors focus only on lncRNAs. True, some lncRNAs might be associated with antimetastatic effect; however, there are many other effects as well.

As the referee suggests, there are molecular alterations of all kind that olaparib may infringe in the tumor cells. We have previously studied the antimetastatic effect of PARP inhibitors in relation to several aspects of tumor biology. However, no studies have been performed so far to analyze its effect on lncRNAs. Based on the increasing interest on the biology of lncRNAs and the need to get further insights in their role in cancer progression we focus this study in this specific type of RNA.

5: In addition, the authors did not really address the previous criticisms, as well.

As previously explained in this response and in the previous version, we have tried to answer to every criticism raised by this referee in the most rigorous way and counting on the scientific references available for this purpose.

In the most recent version, we have included a few paragraphs concerning the possibility of passenger effects in the sections discussing the co-occurrence with alterations in tumor-suppressor genes:

“While the illustrated correlations are statistically significant, predicting their biological relevance with bioinformatic tools is not a simple task. As stated above, lncRNA tend to accumulate passenger alterations due to their large size. Assessing whether the co-occurrence of these alterations with the alterations in TP53, IDH1, PTEN and VHL has a biological significance would require further research which, while not within the scope of the present work, represents an interesting niche for future studies.”

“Further research should be carried out in order to elucidate the biological significance of these correlations. As discussed below, these co-occurrences might be passenger effects due to the higher incidence of these alterations in certain tumor types.”

We have based our report in a number of bioinformatic analysis and results which have a statistical significance, and which we consider relevant for the better knowledge and understanding of PARP and PARP inhibitors. A number of PARP inhibitors are currently approved and in use for the treatment of cancer patients, so we deem it important to explore in detail the role that these drugs can potentially play in every aspect of cancer biology. In this sense, and to our knowledge, ours is the first report to evaluate the impact of a PARP inhibitor in the expression of lncRNA, giving as much insight as whole-transcriptome profiling and bioinformatic tools can provide.